# Synthesis, Characterization, Properties, and Biomedical Application of Chitosan-Based Hydrogels

**DOI:** 10.3390/polym15112482

**Published:** 2023-05-27

**Authors:** Ruixi Ye, Siyu Liu, Wenkai Zhu, Yurong Li, Long Huang, Guozheng Zhang, Yeshun Zhang

**Affiliations:** 1College of Biotechnology, Jiangsu University of Science and Technology, Zhenjiang 212100, China; yerx2023@163.com (R.Y.); l18262889331@163.com (S.L.); kevin021128@163.com (W.Z.); 15298355779@163.com (Y.L.); zgzsri@163.com (G.Z.); 2Key Laboratory of Silkworm and Mulberry Genetic Improvement, Ministry of Agriculture and Rural Affairs, Sericultural Research Institute, Chinese Academy of Agricultural Sciences, Zhenjiang 212100, China; 3Key Laboratory of Biomedical Polymers of Ministry of Education & Department of Chemistry, Wuhan University, 299 Bayi Road, Wuhan 430072, China; longhuang@whu.edu.cn; 4Zhenjiang Zhongnong Biotechnology Co., Ltd., Zhenjiang 212121, China

**Keywords:** biomedical application, characterization methods, chitosan, hydrogel, properties, stimuli-responsive hydrogels, synthesis methods

## Abstract

The prospective applications of chitosan-based hydrogels (CBHs), a category of biocompatible and biodegradable materials, in biomedical disciplines such as tissue engineering, wound healing, drug delivery, and biosensing have garnered great interest. The synthesis and characterization processes used to create CBHs play a significant role in determining their characteristics and effectiveness. The qualities of CBHs might be greatly influenced by tailoring the manufacturing method to get certain traits, including porosity, swelling, mechanical strength, and bioactivity. Additionally, characterization methods aid in gaining access to the microstructures and properties of CBHs. Herein, this review provides a comprehensive assessment of the state-of-the-art with a focus on the affiliation between particular properties and domains in biomedicine. Moreover, this review highlights the beneficial properties and wide application of stimuli-responsive CBHs. The main obstacles and prospects for the future of CBH development for biomedical applications are also covered in this review.

## 1. Introduction

Chitosan is a linear polysaccharide that is the outcome of the partial deacetylation of chitin [1]. Numerous chitosan resources are stored in the exoskeletons of insects and crustaceans (e.g., shellfish and crabs) as well as fungal cell walls [2], meaning they are effortlessly obtainable. Common sources of chitosan and the structures of chitin and chitosan are shown in Figure 1.

The physicochemical properties of chitosan are influenced by its molecular weight (MW), the degree of deacetylation (DD), and the presence of grafted groups [3]. The intrinsically available groups of chitosan encompass C_2_-NH_2_, C_3_-OH, and C_6_-OH [4], so chitosan is a sort of rather active biopolymer that could be altered, activated, and crosslinked via these functional groups. Modifications of this polymer could be performed without affecting the degree of polymerization of chitosan [5]. The major techniques embrace acylation modification, alkylation modification, carboxyl modification, quaternary ammonium modification, etc. [6]. These modifications confer favorable properties and facilitate their application in tissue engineering, drug delivery systems (DDSs), wound healing, biocompatible auxiliary units (BAUs), biosensors, etc.

CBHs are one category of hydrogels based on chitosan, combining the virtues of chitosan and hydrogel, and have wide expenditure. Large volumes of water could be absorbed and retained by hydrogel, a three-dimensional (3D) crosslinked polymer network. Nowadays, smart and multi-functional materials have made great achievements by integrating 4D printing (4DP) and shape memory polymer composites (SMPCs). The 4DP technology can show the characteristic of shape-transforming under the effect of mechanical, chemical, or biological stimuli and has huge advantages including programmed functionality, reduced fabrication time, and a material-saving approach. Smart materials can be divided into two types. One is shape-changing materials (SCMs), and the other is shape-memory materials (SMMs). With the help of 4DP, shape-memory CBHs are easier to prepare [7]. CBHs made using 4DP are more beneficial for industrial applications and suitable for tailor-made applications [8]. The studies concerning the possible benefits of CBHs in biomedicine have been summarized. This review combines the characteristics of CBHs with their biomedical applications in addition to outlining the synthesis and characterization techniques for CBHs. The diverse applications of smart hydrogel, also known as stimuli-responsive hydrogel, which can respond to external stimuli, have been sorted out. Additionally, development prospects, future perspectives, and existing obstacles of CBHs are discussed.

## 2. Synthesis Methods of CBHs

Diverse synthesis methods would primarily endow traits with divergence. Depending on the ways in which bonding networks are formed, CBHs could be divided into physical hydrogels and chemical hydrogels [9]. In physical hydrogels, the polymeric chains of the hydrogels are bound together by molecular entanglement or secondary interactions (mainly consisting of hydrogen bonding interactions, ionic interactions, and hydrophobic interactions), and the physically crosslinked hydrogels are reversible and restorable, which is advantageous to smart CBHs [10]. Moreover, because of the reversible bonds, physically crosslinked hydrogels are more accessible to acquire the property of self-healing [11]. Generally, the hydrogels formed by physical crosslinking have good biocompatibility without adding any crosslinking agents. Nonetheless, some drawbacks, including poor mechanical properties and the difficulty of controlling their average pore diameters, are inevitable in this kind of hydrogel system. Comparatively speaking, the polymeric chains of chemically crosslinked hydrogels are held together via irreversible covalent bonds. Normally, chemically crosslinked hydrogels possess better mechanical properties and stability due to the stronger chemical bonds or links, but the snag is that most of the crosslinking agents utilized for chemical crosslinking are toxic in vivo, reducing biocompatibility. The main synthesis mechanisms have been summarized in Figure 2.

### 2.1. Physical Crosslinking CBHs

There are two prerequisites that must be met in order for a molecular network to have the properties of a hydrogel: (1) the interchain interactions must be strong enough to form a stable structure in the network, and (2) the network must make it simple for water molecules to enter and persist there. The physically crosslinked CBH is rather fragile due to the unstable bonds, but the secondary bonds could be exploited to fabricate CBHs to fulfill these criteria [12]. Since chitosan has protonated amino groups under specific circumstances, it can form ionic complexes through ionic interactions with negatively charged molecules and between anions. Based on electrostatic interaction, chitosan can form ionic complexes with small anionic molecules (e.g., sulfates, citrates, and phosphates) or metal anions or ions in mixed-charge systems [13,14]. Other secondary interchain contacts, such as hydrogen bonds between the hydroxyl groups of chitosan molecules and ionic molecules or connections between deacetylated chitosan chains after cation charge neutralization, are also present during the ionic complexation [15]. In addition, polyelectrolytes are macromolecules with a wide range of MWs, and the direct bonding between chitosan polymers and polyelectrolytes is stronger than hydrogen bonding or van der Waals interactions. Water-soluble anionic macromolecules such as deoxyribonucleic acid (DNA), anionic polysaccharides (e.g., alginate, carboxymethylcellulose, pectin, dextrose sulfate, xanthan gum, etc.), proteins (e.g., gelatin, albumin, sericin, keratin, and collagen), and anionic synthetic polymers (e.g., polyacrylic acid) are inset to fabricate polyelectrolyte CBHs. The stability of these compounds depends on their charge density, solvent, ionic strength, pH level, and temperature [16,17]. Additionally, hydrogels can be formed by polymer blends between chitosan and several polymers that can form hydrogen bonds (e.g., polyvinyl alcohol (PVA)). This method is called freeze-thaw, which involves mixing polymers in an aqueous solution, freezing them at a low temperature, thawing them at room temperature, and conducting continuous cycles to promote physical interactions between polymer chains. Chain-link interactions are the cross-linking sites [16,18]. In the case of chitosan-PVA polymer blends, increasing the chitosan content negatively affects the formation of PVA crystals, leading to the formation of structurally disordered hydrogels [12]. Moreover, chitosan could also be prepared as CBH alone without the addition of any other polymers or complexing molecules, i.e., chitosan can be self-crosslinked when the initial polymer concentration exceeds the critical concentration of chain entanglement and when the hydrophile interactions reach equilibrium [10].

### 2.2. Chemical Crosslinking CBHs

Chemical crosslinkers are molecules with two or more reactive ends that can chemically bond to particular functional groups on proteins or other molecules, such as primary amines, sulfhydryls, etc. Chemically crosslinked CBHs are mainly fabricated by using chemical crosslinking agents to react polymer functional groups with crosslinking agents to cross-link molecules into the 3D scaffold by covalent or ligand bonds. The crosslinker acts as a bridge connecting different or identical polymer chains to form a 3D network, with the mechanical strength and chemical stability of the polymeric material improved [19]. The simplest crosslinks are usually synthesized by condensation of the amino group of chitosan with the carbonyl group of the aldehyde or ketone by elimination of water molecules [20], e.g., dialdehydes, especially glutaraldehyde, could form covalent imine bonds with the amino groups of chitosan by the Schiff reaction. The formation of dynamic covalent bonds results from the Schiff base reaction and endows hydrogels with self-healing properties [18]. The most commonly used crosslinkers for the preparation of CBHs are epichlorohydrin (ECH), ethylene glycol diglycidylate (EGDE), GLA, and genipin [18,21]. The gelation pathway has been listed in Figure 2. The cytotoxicity of crosslinking agents and their tendency to cause inflammatory reactions in vivo are the main obstacles. Genipin is widely utilized as a crosslinking agent due to its low toxicity, making it a fabulous crosslinking agent for CBH preparation [22]. Under acidic and neutral conditions, genipin reacts spontaneously with primary amines on the polymerization chain. Nonetheless, a large quantity of free amino groups would be expended during the preparation phase, reducing its ligand density and reactivity in subsequent reactions. Therefore, an interesting way to impart specific properties to chitosan is to modify it by accessing various new functional groups (such as amino, sulfur, phenolic, and other groups) or molecules (such as acrylic acid), known as grafting. The monomers to be grafted can be single or multiple; they act as side links to chitosan through covalent interaction. This method does not disturb the initial backbone, thus maintaining the basic properties of chitosan [23]. Photosensitive functional groups could be grafted to form polymer blends of hydrogels in situ; chitosan photo-crosslinking precursors are usually obtained by methacrylic acidification, and the polymers can be crosslinked by forming random hydrogel networks under irradiation such as ultraviolet, X-ray, or γ-rays [24]. The advantages of this technique over conventional chemical methods are ease of formation, speed, safety, and low cost. The γ-ray irradiation method for the preparation of chitosan/gelatin/PVA hydrogels improved the tensile strength of chitosan/gelatin/PVA hydrogels compared with gelatin/PVA hydrogels [25].

## 3. Characterization Methods

A number of common characterization methods have been used to characterize CBHs, depending on their different qualities and properties. In general, the microstructure, chemical interactions, thermal stability, biocompatibility, mechanical resistance, antimicrobial properties, and viscosities of the CBHs are key factors to consider [26]. CBHs often require performance testing and structural investigations because of their wide range of applications. The performance properties of CBHs have huge differences depending on their specific applications.

### 3.1. Microstructure Analysis

Numerous investigations of microstructural properties that may have an impact on the structural integrity of CBHs are continuously being carried out. Scanning electron microscopy (SEM) and transmission electron microscopy (TEM) are commonly used [27,28] as microscopic methods for direct and reliable imaging of CBHs. SEM is an intermediate observation between transmission electron microscopy and light microscopy, while TEM projects an accelerated and aggregated electron beam onto a very thin sample, where the electrons collide with the atoms in the sample and change direction, resulting in stereo angular scattering. The size of the scattering angle is related to the density and thickness of the sample, so that different images of light and dark can be formed, which will be displayed on the imaging device after magnification and focusing. Benefiting from the micro-porous structure of CBHs, which could lead to high internal surface areas with low diffusional resistance, the cumulative nystatin in vitro will release after three months from the CBHs [29]. Kocak et al. [30] examined the synergetic effect of glycerol and pH factors on diverse properties of CBHs, and they analyzed microstructural data with SEM in detail. The findings demonstrated that all CBHs displayed a morphology with a flat, smooth bottom surface and a rougher, more numerous top surfaces, with the majority of the pores located at cross-sectional areas, and the pore structure was more uniform in the group with a lower pH level. From the observed SEM images, it could be found that in both initial and modified CBHs, the increasing amount of the crosslinking agent used during the synthesis process caused an increase in the crosslinking density of CBHs. Additionally, the patent influence of the modifier on the CBHs’ surface morphology can also be observed [31].

### 3.2. Chemical Interactions Analysis

It is widely acknowledged that spectroscopic methods are often utilized to test the intrinsic chemical connections between the functional groups that make up the structure of hydrogels. When preparing hydrogels, Fourier transform infrared spectroscopy (FTIR) is widely used in order to surveil the progress [32]. FTIR is the mathematical processing of the Fourier transform by computer technology and its analysis and identification by infrared spectroscopy. It has been used to determine the functional groups of unknown substances and their chemical structure, observe the chemical reaction process, distinguish isomers, and analyze the purity of substances. For instance, Bańkosz et al. [33] observed an increase in the intensity of this band in the CBH modified with albumin particles by FTIR, indicating the presence of the above-mentioned modifier, i.e., albumin particles, in the tested material. In the preparation of CBH using graft copolymerization, amino groups from chitosan were confirmed to be involved in the grafting reaction by FTIR [34]. Moreover, N-H and O-H stretching vibrations can also indicate chitosan has participated in crosslinking during the CBH formation process [35].

### 3.3. Thermal Stability Analysis

Using thermogravimetric analysis, it is possible to determine the thermal and oxidative stability of the material under different atmospheric conditions, analyze the physicochemical processes of decomposition, adsorption, desorption, oxidation, and reduction of the material, including further apparent reaction kinetic studies using the results of thermogravimetry analysis (TGA) [36], quantify the composition of the substance, and determine the content of moisture, volatile components, and various additives and fillers. TGA can be used to evaluate the thermal properties of the aryl-functional group-crosslinked CBHs they generated. It was demonstrated that the final degradation temperature of all derivatives was lower than that of chitosan due to derivatization and the thermal stability of the chitosan derivatives was worse than that of chitosan. This instability could be attributed to the deterioration of the crystallinity of chitosan due to the formation of Schiff bases [37].

### 3.4. Biocompatibility Analysis

The hemolysis rate is one of the most significant evaluation criteria when taking materials in blood-material contact for wound healing into consideration [38]. The hemolytic reaction determines the degree of hemoglobin release caused by materials. Other particular hemocompatibility examinations can also be designed to simulate the shape and flow dynamics of medical devices or materials during actual application and to determine the interaction between blood, material, and device. The hemolysis rate is calculated using the equation:hemolysis rate [%] = ([OD]specimen − [OD]negative)/([OD]positive − [OD]negative) × 100%,(1)
where [OD]specimen is the absorbance for samples, [OD]negative is the absorbance for the negative control, and [OD]positive is the absorbance for the positive control.

For example, the hemolysis rate for chitosan/glyoxal hydrogels without washing or immersion in a polyphenol solution was 63.84%, which means that such hydrogels are highly hemolytic. Nonetheless, the hemolysis rate was negative after washing with distilled water. By washing the hydrogels with water, unreacted glyoxal is removed, and the rate of hemolysis is reduced [39].

### 3.5. Mechanical Resistance Analysis

The mechanical resistance of CBHs is regarded as an important quality for their application in various scenarios. CBHs can be used as scaffolds in drug delivery and diverse biological applications. The CBHs are supposed to possess enough mechanical resistance to maintain structural integrity. The mechanical resistance of CBHs is usually estimated by cutting the materials into specific shapes and examining their abilities to be elongated and tensiled using a universal testing machine, while the thickness of the CBHs is measured with a caliper. Furthermore, the puncture test can also be used to evaluate the mechanical resistance of CBHs [40]. However, one of the most marked disadvantages of CBHs is that the vast majority of their mechanical resistance is poor. The multi-network structure is a splendid solution for increasing the mechanical resistance of CBHs, which is shown in Figure 3a [41]. In addition, CBHs combined with electrospinning can also exhibit sufficient strength and flexibility; the microstructure could be visualized via the SEM technique [42].

### 3.6. Antimicrobial Properties Analysis

Chitosan has been shown to inhibit the growth of bacteria, filamentous fungi, and yeast strains. Chitosan has also been found to be an antimicrobial agent. To characterize the antimicrobial properties of CBHs, the inhibition ring method [43], the OD counting method [44], and the SEM method [45] have been applied. When comparing these methods, several factors should be considered, including sensitivity, ease of use, cost, and the type of data obtained (qualitative vs. quantitative). Among the three methods, the inhibition zone method is convenient, simple, low-cost, accurate, and reliable, while the OD counting method has a simple operation and only a few simple steps are required. The choice of method depends on the specific research objectives, available resources, and desired level of detail in the antibacterial assessment of hydrogels. Among the developed CBHs, the carbomer 940 (CBM)/carboxymethyl chitosan (CMC)/Eucalyptus essential oil (EEO) hydrogel exhibited optimal antibacterial activities of 46.26  ±  2.22% and 63.05  ±  0.99% against *Staphylococcus aureus* and *E. coli*, respectively, along with cell viability (>92.37%) and migration activity [46].

### 3.7. Swelling Analysis

The swelling test is an evaluation criterion that is necessary to determine the proportion of biological media that could be retained inside the hydrogel network. The weight of dry hydrogel is measured as a control, and then the dried sample is placed into biological media for about 48 h. Furthermore, the temperature variation can also be used in thermo-sensitive hydrogels. After the swelling procedure, the weight of the water-adsorbed hydrogel is measured to calculate the quality of the water in the dried hydrogel. Due to the large relative MW and polydispersion of polymers, the dissolution phenomenon of polymers is much more complex than that of small molecule compounds. The swelling ability of the hydrogels was defined using the swelling ratio (α), which was calculated by the following equation:α = (m − m_0_)/m_0_,(2)
where α is the swelling ratio, *g/g*; m is the mass of swollen hydrogel, *g*; and m_0_ is the mass of dry hydrogel.

For instance, Kudłacik-Kramarczyk et al. [47] characterized the sorption properties of both the unmodified CBHs and the CBHs containing albumin, and they demonstrated that while the swelling ratios of the modified polymers were slightly greater than those of the modified CBHs, the discrepancies between the values of the swelling ratios estimated for the modified CBHs and the CBHs without albumin were minimal. The hydrogel viscosity is mostly tested with sol-gel transition measurements. The test is necessary for the research and development of thermosensitive CBHs. When stimulated by temperature changes, thermosensitive CBHs transform from the solution state to the gelation state [48].

## 4. Properties and Biomedical Application of CBHs

CBHs are incredible biomaterials with an extensive range of interesting applications. Most CBHs inherit the essential property of benign biocompatibility from chitosan [49]. The marvelous capacity to be characterized, meanwhile, allows the widespread application of CBHs in biomedicine. The characteristics and applications of one hydrogel are generally regarded as being closely connected. Herein, the main properties of CBHs catering to versatile application sectors are methodically introduced. This section provides a thorough overview of the current state-of-the-art and prospects in biomedicine. CBHs could be endowed with versatile properties such as swelling ability, stimuli-responsiveness, antimicrobial activity, enhanced mechanical strength, etc. Consequently, CBHs could be applied in a diversity of biomedical sectors, including tissue engineering, DDSs, wound healing, etc., in accordance with their properties, as shown in Figure 4.

### 4.1. Tissue Engineering

Acceleration or augmentation of regeneration procedures is a requisite in tissue engineering due to the insufficient ability of some target tissues to propagate. For instance, cartilage is a form of connective tissue that has a low capability for self-repair due to its avascular and aneural nature [50]. Therefore, one feasible treatment method is to foster regeneration via exotic hydrogels. Hydrogel is a promising material for tissue engineering since its 3D configuration is analogous to the natural extracellular matrix (ECM) of tissues, while the ferous porosity arrangement supports cell adhesion, proliferation, differentiation, and function. In addition, surface modification with other substances can foster the step [51]. Gifted properties facilitate the preparation steps of the scaffold, but the successive activities are tricky, so these are detailed.

#### 4.1.1. Methods and Techniques to Implant

The scaffold is often fabricated in a laboratory setting before being surgically implanted into the flaw site [52]. This technique incorporates materials or cells into the hydrogel and allows for fine control over the size and shape of the hydrogel. However, it might also impair surrounding tissues or cause an infection or inflammation. 3D bioprinting is an advanced fabrication technique that can produce patient-specific scaffolds with complex geometries and precise control of cell positioning and can achieve large tissue-engineered products, with shear thinning behavior and viscosity being the critical characteristics of the bioink [53,54]. The 3D printing (3DP) technology could overcome limitations in complicated tissue conditions with unique forms [55]. Chitosan is a high MW polymer that produces viscous solutions for 3DP. This technique has the advantages of creating homogenous pore sizes and avoiding deviation compared with conventional CBH construction methods [56]. Xu et al. [57] reported a supramolecular CBH with host-guest connections that had great printability and mechanical strength. Comparatively, in situ tissue engineering avoids the need for pre-culturing cells and tissues in vitro and uses the host’s own cells and biomolecules to regenerate the tissue, improving the integration and functionality of the regenerated tissue while lowering the risk of immunological rejection and infection [58]. In situ-forming CBH, which is a kind of smart hydrogel that is able to commence sol-gel transition in response to exterior inducements, such as pH value and temperature [59], could be injected into the body via minimally invasive surgery (MIS). Chitosan is water-soluble and positively charged under acidic conditions (pH < 6), but would turn uncharged and hydrophobic at physiological pH, forming a compact and physically crosslinked hydrogel [60]. The generation of both parallel and perpendicular crosslinking in CBH network structure arises at higher pH levels (above 6.5) (tissue pH is neutral) as well [61]. The two mechanisms facilitate pH in situ formation. Though mechanical properties would descend unavoidably, concentration and printing temperature are vital parameters affecting the gelation rate and thereby the strength of printed structures in in situ gelation four dimensional (4D) CBH, which is endowed with thermo-sensitivity [62]. Introduced thermo-responsive constituent parts enable CBHs to respond to temperature via the quantity variance of intramolecular hydrogen bounds, and this has been considered a typical stimulus for smart CBHs [63]. Nevertheless, before application, it is vital to take into account patient variability, hydrogel displacement, lack of stability, etc.

#### 4.1.2. Microenvironment Adjustment

A nonthreatening microenvironment is critical for tissue regeneration. There are two dimensions to improving conditions: one is to eliminate antigens that might trigger an immunoreaction, and the other is to modulate the host immune response. Bioactive materials, e.g., growth factors, peptides, nucleic acids, and antibiotics, could be loaded by CBHs to facilitate the growth of seed cells. Growth factors can be chemically modified or covalently immobilized via the amino and hydroxyl groups of chitosan, improving their stability and bioactivity in the target tissue matrix [64]. Li et al. [65] reported that CBH could recruit tetrahedral framework nucleic acid (TFNA), which was injected into the articular cavity to enhance cartilage repair. TFNA is a promising DNA nanomaterial for improving the regenerative microenvironment. The antibiotics added to the hydrogel can enhance antimicrobial activity, producing environments that are free of microbes [66]. Inflammation resulting from xenografts is one obstacle to the implantation of CBHs [67]. Therefore, immunomodulation via hydrogel becomes conspicuous. Disordered macrophage activation impedes tissue regeneration. Regulating macrophages from the M1 type to the M2 type is crucial for expediting tissue repair in cartilage [68] and bone regeneration [69] because the M1 type promotes inflammation while the M2 type secretes anti-inflammatory cytokines for pro-tissue repair effects. Various CBHs could regulate this procedure. For instance, chitosan/silk fibroin/cellulose nanoparticle (CS/SF/CNPs) scaffolds facilitate M2 macrophage polarization and influence the osteo-immunomodulatory responses of the cells [69].

#### 4.1.3. Tissue Regeneration

Cells interact with their microenvironments and are affected by them. Hydrogels must be created with exclusive physical, chemical, and biological qualities in order to account for the unique traits of each type of tissue and foster tissue regeneration.

Regeneration strategies can be divided into two types: encapsulating autologous cells as seed cells in hydrogels and inducing the proliferation of surrounding cells via various substances [50]. The combination of cell-based and proliferative strategies is ideal. In the experiment of Hao et al. [70], CBHs alone were less effective than chondrocyte- and chitosan-coated materials at fully repairing cartilage lesions in vivo, proving the advantages of adding seed cells. Similar to the natural ECM, the matrix stiffness of CBHs is one parameter affecting tissue regeneration by regulating the force exerted on the cells. It has been demonstrated that the hydrogel matrix stiffness affects cell proliferation, with early passage-stage cells being more sensitive [71]. High matrix stiffness typically encourages cell growth, whereas low stiffness causes cell dormancy and stemness [72]. The ECM stiffness varies depending on the tissue type and can influence cell behavior, so modulation is imperative for application. Generally, the parameter can be adjusted by modifying the crosslinking degree; more bonds are beneficial to enhance the deformation resistance, while fewer bonds work in the opposite direction. Chang et al. [73] achieve stiffness tunability by changing the grafting ratio. Dual-crosslinked hydrogels might have higher stiffness than single-crosslinking methods [74]. Moreover, the covalent binding methods unavoidably affect other properties due to the changes in CBH structures, so it turned out to be an optimization problem. Physical contacts are weaker and less stable than dynamic covalent bonds, which are reversible, stimuli-responsive, and adjustable [75], a workable method for controlling the stiffness of hydrogels. However, a few issues with the latter, including achieving homogeneity, incomplete study, and finite reaction type variety, need to be ameliorated [76]. Additionally, the porous structure is available for the proliferation of the tissue because it can not only facilitate the adhesion and proliferation of adjacent cells but also release drugs more efficiently. CBHs could be made permeable using a variety of techniques, divided into physical, chemical, and biological types. The porous framework of CBHs can be generated by 3DP [53], foaming [77], microwave [78], etc. NaHCO_3_, added to the CBH system, could react with the protonated amino groups in vivo, releasing CO_2_ to generate micropores [79]. Adding lysozyme encourages the degradation of CBHs to generate pores. By catalyzing the breakdown of CBH, lysozyme can be used to produce porous structures. When mesenchymal stem cells are incorporated into chitosan-lysozyme hydrogels, these spaces advance cell proliferation and migration, which likewise help with osteogenic differentiation [80]. The favorable average diameter of scaffold pores varies depending on the cell type, which has an impact on cell differentiation and gene expression [81], so regulating the average diameter of the hydrogel is vital, with the modifying approaches diverging between techniques. Furthermore, the porosity ratio is another critical parameter to assess the microstructure. In addition, 4D CBH, a type of smart CBH based on the 3DP technique [82], holds the potential to manufacture sophisticated non-sintering scaffolds with improved porosity qualities for hard tissue engineering scaffolds or mimic the ECM of soft ones to foster tissue regeneration [83]. Nonetheless, Plama et al. [84] evaluated the tissue regeneration histologically and found CBHs worked with no extraordinary results compared with conventional treatments for tissue regeneration in immature teeth. In addition, they claimed the CBHs used in the tests had drawbacks such as insufficient degradation, ongoing inflammation, and a lack of mechanical support for the coronal seal. These unsatisfactory outcomes might be attributed to the crosslinking methods, DD, MW, etc., of CBHs. Therefore, the correlation between functional ingredients or parameters of CBHs and their potential biomedical applications deserves further research.

#### 4.1.4. Biodegradation In Vivo

As tissue engineering materials, scaffolds should not induce acute or chronic effects and be biodegradable because the newly formed tissue should be able to replace them simultaneously [85], which prevents the need for surgical removal. Hydrolysis, enzyme-mediated processes, or a combination are utilized [60]. Lysozyme, a critical enzyme catalyzing the degradation of CBH, resides in all mammalian tissues and functions via accidental splitting of β-1,4-glycosidic bonds (depolymerization) and hydrolysis of the N-acetyl linkage (deacetylation) [85]. It has been proven that the degradation of CBH can be tweaked by changing the concentration of lysozyme added to chitosan hydrogels by chemically altering methacrylate groups. Additionally, biodegradable smart CBHs are hydrogels that would break down over time in response to biological or microenvironmental ingredients such as biological substances, pH levels, temperature, etc. Biodegradable smart hydrogels have been utilized for tissue engineering purposes because of their advantages, including controlled degradation kinetics and sensitivity to incentives. Nevertheless, some of the challenges or limitations are their low mechanical strength, inferior stability, multifactorial degradation kinetics, and potential immunogenicity [86]. Notably, CBHs can release loaded drugs, cells, growth factors, etc., following degradation, and the released N-acetyl-β-D-glucosamine induces fibroblast growth [87]. There are several parameters in the synthesis procedure that influence the biodegradability, such as the DD and MW of chitosan and the type and concentration of crosslinking agents [10]. Nonetheless, the impact of the concentration of lysozyme in the tissue, which will ascend in some adverse cases, on the biodegradation ratio is still unstudied, which should be considered in the factors affecting it. Last but not least, Reay et al. [88] measured the sizes of the genipin-chitosan hydrogel-degraded particles and reported that the majority of them were as small as 1.7 nm, which is below the renal filtration threshold, meaning that they could be eliminated from the body via urine. The safety of CBH biodegradation remains unclear due to the varied crosslinking methods of CBHs; therefore, further research is crucial.

### 4.2. Drug Delivery Systems (DDSs)

CBHs have been manipulated as drug delivery vehicles for various bioactive agents. CBHs could enhance the stability, retention, and bioactivity of the delivered agents and modulate the release kinetics and interactions with the target tissues. Moreover, many medicines with different physicochemical properties could be loaded by the CBH to achieve treatment.

#### 4.2.1. Drug Release Regulation and Release Kinetics

Stimuli-responsive CBHs are the major component of DDSs made of CBHs, having been well-researched and widely applied, for they are liable to gelatinize and break down under definite stimuli. In addition, their widespread application can be attributed to their injectability, lack of surgical necessity, shape flexibility, etc. They can release drugs, cells, biomass, etc., at one specific occasion, depending on the microenvironment of the tissue, including pH values, temperature, biological factors, etc. [89]. Passively received and steerable stimuli could be used to categorize these signals. The former should be nimble and take into account the hardwired circumstances of the target site, due to the fact that the former is a category of indeterminate parameters, whereas the latter can be regulated because it is generally stable and mostly factitious. As detailed in the tissue engineering part, at an acidic pH, the amino groups of chitosan will be positively charged, and the hydrogel will swell, facilitating the release of cargo. However, most of the groups were depleted by chemical gelation. Silva et al. [90] preserved the original amino acid via a protecting and deprotecting strategy. The physical crosslinking approach is another practical option. It can retain relatively high pH sensitivity and fragility, which are advantageous for the emancipation of medications. It has been demonstrated that sol-gel crosslinking CBH is a pH-responsive matrix that can bulge or dwindle depending on the pH level of the microenvironment [91]. In cancer therapy, the drugs used in DDSs are usually malignant to normal cells as well. The gel formation is rather immobile, favoring adhesion to the targeted tissue by filling it up or wrapping it. Then, different stimuli foster the release of bioactive molecules integrated with chitosan, achieving targeted therapy. The tumor microenvironment (TME) is a multifaceted and continuously evolving object, featuring acidic, hypoxic, excessive metabolite accumulation, and high-depression conditions for immune cells [92,93]. pH-responsive CBHs, especially acid-sensitive ones, would trigger degradation of the hydrogel there [94], thereby releasing the loaded drugs. Another latent reaction, swelling behavior, triggered by acidic stimuli, increases the contact area between the hydrogel and the tumor, facilitating the diffusion of drugs or other agents from the hydrogel [89]. In addition, it can also exert mechanical pressure on the tumor tissue, depressing and hindering tumor growth. The high density of lactic acid in the tissue fluid promotes macrophage transformation into the M2 type, which will foster the escalation of tumors [95]. The immunosuppressive milieu can be reversed in the microenvironment by adding CaCO_3_, which also lessens the immunosuppressive effect on T cells since the higher pH value encourages macrophage polarization from the M2 type to the M1 type [96]. Additionally, the bionic onion structure is applied to regulate the drug release kinetics [97]. It enlightens that the exquisite structure can be programmed to release alkaline salts or ions in acidic instances before setting free medications, which are not only stimuli-responsive but also can adjust the microenvironment to favor subsequent treatment. Thermosensitive CBHs can be divided into two types. One form of thermosensitive CBH exhibits a lower critical solution temperature (LCST), whereas the other type exhibits an upper critical solution temperature (UCST). When the temperature is raised over a certain degree, the LCST hydrogel contracts and releases contents, whereas the UCST hydrogel expands and absorbs water. LCST could be exploited to control the release of drugs from the hydrogel, for the hydrogel will shrink and release encapsulated drugs within it [98]. By controlling the temperature and the LCST of the CBHs, it is achievable to control the rate and timing of drug release. Comparatively, the delivery mode of in situ-forming hydrogels is the very opposite of the anterior. UCST hydrogel is a fluid that facilitates a mixture with drugs at a lower temperature and becomes a gel after being inoculated into the body. Incidentally, the UCST CBH is widely utilized as a category of injectable hydrogels. The liquid formation facilitates the homogeneous incorporation of drugs or biological molecules, and they can keep their hydrogel formation after injection, achieving relatively sustained drug release. Vaccine antigens could be added when the hydrogel is liquid at room temperature and can be released slowly to trigger the adaptive immune system [99]. Wu et al. [100] creatively synthesized hydrogels via ascending the temperature break of the LCST; this is promising to prepare DDSs. Hydrogels that are light-sensitive have a variety of methods to release their cargo. One method involves the photothermal effect, in which near-infrared light can cause a specific cargo to be released, leading to the creation of a hydrogel [101]. The release of cargo can also occur through photo-initiated chemical processes, which could alter the hydrogel network [102]. Nevertheless, these CBHs only respond to one parameter, and it is vital to control or maintain other conditions during the preparation and application phases, limiting their potential applications and controlling their behavior. Multi-responsive CBHs are hydrogels that can respond to more than one stimulus from the environment. They can control drug release precisely and lessen side effects thanks to their benefits, including diverse sensitivities, expected kinetics, easier degradability under particular conditions, etc. [103]. Nisar et al. [104] synthesized an ultraviolet (UV)- and pH-sensitive ONB-chitosan hydrogel via the Schiff base reaction, forming imine bonds. Nonetheless, the synthesis procedure is complicated and more likely to retain harmful chemical agents. Theoretically, because physical crosslinking CBHs are fragile structures, it is possible to create them, excluding the latter hidden problem, but it is challenging to manage how they degrade in response to particular stimuli. The swift release mechanism is appropriate for rapid onset or potent drugs. The opposite can release with a relatively average velocity and preserve stability under some conditions, which is preferable to drugs that only need to maintain low consistency and a sustained release over a longer period of time. Furthermore, by avoiding the high peak density of medications in the body, sustained release can also aid in lessening adverse effects and damage to contiguous tissues. Notably, 4D CBH could undergo deformation under specific stimuli that suit the task due to their degradability to release loaded drugs in a controlled way, and they can be produced into customized and controlled DDSs [82,105]. The stimuli could trigger the unfolding, swelling, deformation, or degradation of the 4D material [54], which can be used for reference to endow 4D CBH with release drugs at specific sites and times. 4D bioprinting can overcome some of the limitations of conventional drug delivery systems, such as bad responsiveness, low bioavailability, low recovery, etc. [105]. Except for sensitive CBHs, drug release kinetics could also be tunable via multi-network [86,87] CBHs, self-digestion, hydrolysis, etc. Lysozyme can act as a crosslinker for chitosan sulfate hydrogels, and the release kinetics of drugs can be modified by adjusting the lysozyme concentration [88]. The hydrolysis of the azomethine linkages in the polyurethane chains led to the degradation of the hybrid hydrogels [106].

#### 4.2.2. Loaded Medicines

CBHs could perform as DDSs for medications by incorporating or adsorbing them within their network or via covalent bonds. The merits of CBHs as delivery carriers include versatile loading modes, extending the retention of consignments in biological settings, and regulated or controllable release kinetics [107]. Hydrogel is a 3D structure with a high water content, so hydrophilic medications are more likely to load than hydrophobic ones. Considering the increasing needs, there are several strategies to increase the load ratio and retard the release of hydrophobic drugs. It has been demonstrated that boosting the proportion of hydrophobic moieties inside CBH can postpone the release, but the link should be correctly regulated because the relationship is not a linear one [108]. Polymeric micelles, self-assembled configurations of amphiphilic molecules having a hydrophobic core and a hydrophilic shell, can also be exploited to extraordinarily depress diffusion of the hydrophobic substances [109]. A biocompatible, non-ionic microemulsion can be used to encapsulate hydrophobic drugs to enhance the water solubility of the drugs and protect them from degradation [110]. Conspicuously, encapsulating exosomes and micro-vesicles, two types of extracellular vesicles that are small membrane-bound structures released by cells into the extracellular space, are capable of carrying different types of macromolecules, bestowing CBHs with the aptitude to modulate several metabolic pathways in the recipient cells [111,112]. The amalgamation of the two materials is favorable because they have the advantages of being stimuli-responsive, biocompatible, able to steadily release, able to incorporate a variety of chemicals, etc.

### 4.3. Wound Healing

The intention of wound healing is to accelerate the natural healing response so that damaged tissues can mend and regenerate. This involves the use of dressings, biomaterials, and drugs that can modulate the hemostasis, inflammation, proliferation, and remodeling phases [113] of wound healing, with CBH functioning mainly in the first three phases. Wound-healing CBHs can be operated on to treat acute wounds, chronic inflammation, ulcers, etc.

#### 4.3.1. Hemostasis Phase

Hemocompatibility is a prerequisite for wound-dressing materials [114], as these CBHs come into direct contact with gore. To improve the adhesion property of hydrogels on bleeding wound surfaces of internal organs, Chen et al. [115] suggested that creating a local drying setting by expelling the moisture on the wound surface via incorporating a super-hydrophilic substance is one feasible way. Furthermore, the hydrogel could still preserve a humid microenvironment, endorsing wound healing [116], inside the wound by absorbing effusion. Introducing other adhesive components can also help [114]. After being injected into irregular wounds, self-healing CBHs are capable of mending on their own thanks to the design of reversible crosslinking, including Schiff base bonds, the metal coordination bond, host-guest interactions, and electrostatic interactions [117]. Wei et al. [118] developed a particular type of CBH that demonstrated a normal shear-induced gel-to-sol transition as well as quick self-healing ability and quickly plugged the lesion with the CBH. Additionally, photo-initiated polymerization allows for the tuning of the CBH, which enables it to respond to various wound forms and tissue types. Moreover, another benefit is that the wound dressings can preserve integrity under external forces. After the hydrogels are applied to the location, versatile methods, such as electrostatic interaction [98] and chitosan’s inherent hemostatic property [119], are implemented to concentrate blood cells and platelets to activate coagulation, the initial stage of wound healing. Yu et al. [120] synthesized Janus self-propelled CBH spheres with constructive swelling behaviors, realizing exudate absorption, red blood cell and platelet concentration, and directional movement, moving against blood flow and reaching the bleeding site. Notably, apart from these methods, the native coagulation system in vivo can also be regulated via loaded substances. Ca^2+^ is widely integrated into CBHs through ionic [114] or calcium salt [121] formation. The ion can induce the conversion of prothrombin to thrombin, a critical step because the latter fosters the formation of fibrin networks [120]. Additionally, tranexamic acid (TXA), a typical antifibrinolytic drug to treat hemophilia and other coagulopathies [122], can be added, avoiding fibrinolysis and preventing the breakdown of blood clots [120]. Song et al. [123] demonstrated that a certain ratio of acetyl amino and amino groups in chitosan may also stimulate the coagulation system and encourage platelet and erythrocyte adherence. However, whether there are similar mechanisms in CBHs deserves further research.

#### 4.3.2. Inflammation Regulation Phase

Bacterial infection is a tricky menace to laceration and escalating inflammation. CBHs that have inherent antibacterial properties and combining antibacterial agents with hydrogels are two viable strategies. The former relies on the intrinsic properties of chitosan (or its derivatives) or is triggered by some substances (e.g., triazole rings [124]). The positively charged amino groups (pH < 6) of chitosan electrostatically interact with negatively charged components on the microbial membrane, thwarting basic roles [125]. Some low-MW chitosan can permeate the membrane and combine the biopolymers, disordering metabolisms [126]. The latter involves the incorporation of external antibacterial agents, such as antibiotics (e.g., lincomycin [127]), metal nanoparticles or ions (e.g., silver nanoparticles (AgNPs) or Mg^2+^), natural polymers (e.g., antimicrobial peptides [128]), etc., into the hydrogel matrix to eliminate microbes due to its biodegradability. Nonetheless, CBHs that have inherent antibacterial properties may have lower and narrower antibacterial activity and spectrum than those combining antibacterial agents with hydrogels. Therefore, both have merits and demerits, and the choice of the most suitable one depends on the specific application and requirements. Inflammation is a normal response to injury or infection, but excessive or chronic inflammation, more common in the type 2 diabetic patient group, would result in tissue damage. Therefore, on the premise that no microorganisms exist or have any growth activity, anti-inflammation is imperative to decrease the damage. There are versatile methods to downregulate it. For instance, by inhibiting the expression of inducible nitric oxide synthase (iNOS), an enzyme that creates NO in response to inflammation, it was demonstrated that the diminution in NO instigated by CBH had an anti-inflammatory impact, activated by lipopolysaccharide [129]. Nevertheless, there are some concerns that the antibacterial agent would leach out over time, resulting in the loss of antibacterial property [130], and the anti-inflammation property might work reversely. Yin et al. [131] synthesized hybrid hydrogels, which could elevate the release of both NO and tumor necrosis factor alpha (TNF-α) by stimulating and activating RAW 264.7 macrophages, augmenting their antibacterial biological response, which is proven to advance the healing procedure. In short, the regulation of inflammation should be a comprehensive issue, demanding consideration of the antimicrobial property, adjacent tissue state, patient condition, etc.

#### 4.3.3. Fostering Proliferation Phase

The third stage is proliferation, where new tissue is formed to rebuild the site, which is detailed in the tissue engineering part. Another way to promote proliferation is to eliminate malignant factors. To a great extent, delayed wound healing results from the depletion of metal ions in the wound microenvironment, so the controlled release sequence of metal ions from CBHs can significantly promote wound healing [132]. Reactive oxygen species (ROS) buildup greater than the antioxidant capacity of cells can obstruct wound healing, especially in cases of chronic and non-healing wounds, whereupon antioxidant treatments are effective in preventing or reducing oxidative stress and improving wound healing [133]. Amino groups of chitosan can be regarded as electron donors to stabilize active oxidizing radicals and further terminate the free radical chain reaction, which endows it as an antioxidant [134], with prolonged N-deacetylation having a stronger antioxidant effect [135]. Phenolic hydroxyl groups [136], magnolol [137], and other oxidants present in the hydrogel matrix can enhance the property.

### 4.4. Biocompatible Auxiliary Units (BAUs)

The BAUs are a specific kind of hydrogel with high biocompatibility, which enables them to function as substitutes or auxiliaries for tissues to achieve healing or symptom relief. Generally, BAUs with biodegradability are beneficial for short-term usage, while the others are better suited for substitution. Furthermore, the latter, known as tissue substitutes, aim to mimic the properties of original tissues rather than serve as scaffolds for tissue regeneration, which sets them apart from tissue engineering. Biocompatibility and low cytotoxicity are essential to alleviating inflammation [138] and damage to adjacent tissues [139].

Contact lenses made of CBH can be used for eyesight correction or therapy. The former requires premium optical properties, which could be achieved by regulating the orientation of the polymer chains via electrodeposition [140], while the latter is more of a DDS. Jiao et al. [138] produced contact lenses made of one CBH endowed with biodegradability, antioxidant activity, and antimicrobial activity, making them a feasible therapy for bacterial keratitis therapy and corneal repair. In addition, its optical properties are similar to commercial ones.

As tissue substitutes, the gels shall be endowed with equal or enhanced mechanical characteristics. The difficulty of preparing a permanent substitute is another issue, and research on how to preserve the integration of hydrogels is insufficient [85]. The physical crosslinking of CBH preserves the stimuli-responsibility and biodegradability of hydrogels; the latter is disadvantageous in this case. With coordination crosslinking between chitosan and nanoparticles [141] or other polymers, the mechanical properties of hydrogels can be enhanced. Dual-physical crosslinking is able to enhance stability, depending on other precursors [142]. The cytotoxicity of chemical agents and the inability to react to stimuli are obstacles, though they foster stronger covalent crosslinking. Exploiting novel agents with higher biocompatibility and lower cytotoxicity is one feasible way. Combining the two approaches is another potential course of action, which would give CBH some intriguing advantages and make it possible to produce hydrogels with better mechanical and structural stability while preserving biocompatibility and biodegradability [143]. Tough and lubricated BAU is a great artificial cartilage that functions as the native tissue [144]. The gel membrane with high toughness, immunomodulatory, and anti-adhesion properties may replace the biological ones such as the dura mater and diaphragm [145]. UCST hydrogels are prone to application due to the in situ formation of these gels, enabling injection in MIS, because they are a liquid solution at room temperature and form a gel at body temperature, avoiding surgery [145,146]. Combining this property with pH sensitivity, CBHs could release drugs in a controlled manner [139], achieving the treatment incidentally. Ni et al. [146] synthesized hydrogels, forming a cushion to lift the lesion area, adhere to the wound, and protect the micro-wound from acidic environments.

### 4.5. Molecular Detection Biosensors

Biosensors, a type of analytical tool, consist of biological elements with physicochemical detectors that perceive and quantify the presence or concentration of particular compounds in a sample. Herein, molecular detection biosensors made of CBHs have been set into immobilization matrixes and responsive units. The former is quantitative, inexpensive, and uncomplicated to use and detect in vitro. Nonetheless, the sampling process is typically uncomfortable for the patients. The latter makes the most of the properties of the CBHs, including their high stiffness, sensitivity, avoidance of surgery, etc., but it necessitates various instrumentation for configuration and rigorous manufacturing standards for products.

#### 4.5.1. Immobilization Matrix

The ability of qualitative biosensors to convert or magnify an unquantifiable signal into a quantifiable one is a crucial feature. One strategy is to utilize the rest of the reactive groups of CBH to graft other bioscopes for transformation. The low limit of detection indicates the sensitivity. Immobilization ability, linear range, and surface area-to-volume ratio are critical parameters. Most immobilization matrixes are made into membrane or layer formation because they have a higher surface area-to-volume ratio, meaning that more biomolecules or nanomaterials can be attached to the hydrogel surface, which can improve the sensitivity, selectivity, and stability of the biosensor [147]. In order to create hydrogels with improved mechanical stability and enzyme adsorption capacity, additional polymers can be grafted onto chitosan [148]. Another method is to combine nanoparticles with chitosan to create nanocomposites that have improved conductivity and sensitivity [147]. An enzyme, antibody, or nucleic acid, for example, interacts with the target substance to produce a signal that the physicochemical detector can detect and quantify. Inserting nucleic acid could be implemented to find the complement sequence that measures the level of gene expression, making it easier to diagnose at the gene level [149]. The electrochemical reaction enzyme can transform the concentration of the target substrate into detectable electric signals received by the electrode [150,151,152]. While it is important to take precautions before detection because otherwise these molecules are easily deactivated or degraded, the detecting circumstances are typically complex and may have an impact on the stability of bioprobes, CBHs, and the bonding between them. Another concern is that the sensitivity is easily affected by the solution conductivity. Therefore, it is necessary to increase the stability and set the calibration. Additionally, another technique uses variations in optical signals brought on by the blend of a detection target and CBHs [153,154]. Partial examples have been summarized in Table 1.

#### 4.5.2. Responsive Units

Ebrahimi et al. [160] reported a method by which the presence and concentration of the enzyme can be indicated by the intensity of the fluorogenic substrate, which is covalently conjugated to chitosan and will be released after degradation. Another strategy is to detect hydrogel conductivity. This method is used in detecting vitality signals, e.g., respiratory movements, heart beating, and motions, by inducing minor deformation, which is enough to change the inherent conductivity of the hydrogel [161,162]. Electrical signals are the common output of this method; they rely on the electrode to sense the current change. Conductivity is critical to the sensitivity of biosensors. Given that the hydrogel itself is not conductive, adding carbon nanotubes [161], salt content [162], metal ions [163], etc., can improve it. In addition to improving wear resistance, oriented 3D networks also give CBH enhanced linearity over a broad strain sensing range [144], increasing accuracy. Moreover, many properties of CBHs facilitate biomedical applications including self-healing and adhesion-elongating duration in vivo [161]. The anti-swelling feature maintains mechanical toughness and strength across a variety of deformations and pressures, guaranteeing the stability and dependability of CBH sensors in a variety of settings and applications [163].

Traditional inorganic sensing devices connect to tissues through physical attachment on the surface, whereas the flexibility of tissues limits their application [161]. All stimuli-responsive hydrogels have the potential to be made into biosensors from the perspective of material design [164]. Research on sensing subtle signals in vivo attempted to start. The promising application direction is utilizing the stimuli-sensitive property of smart CBHs to detect micro-changes in the shapes, conductivity, and swelling ratio of CBHs. Additionally, it can be a kind of signal to detect the parameters in vivo. They can be made very small and biodegradable. However, the obstacles are the detection techniques to sense these changes.

Another potential application is actuators. There are many actuators made of composite materials [55]. For instance, Zhu et al. [165] synthesized one CBH actuator that could respond to ion concentration and achieve programmatic deformation. Nonetheless, these units in existing research are not well-optimized and may not meet the medical application requirements due to their sluggishness, unprecise control, size, energy equipment, etc. Besides, CBH actuators could be easily made by the 4DP technique; the combination of both techniques is promising [166]. If the issues are addressed, these CBH actuators would have wide prospects in biomedicine, including vascular valves, artificial muscles, artificial automatic hearts, vulnerable tissue substitutes, etc.

## 5. The Status of Approved Application of CBH in Biomedicine

Accounting for issues with its source, purity, and immunogenicity, the U.S. Food and Drug Administration (FDA) has permitted the use of chitosan as a biomaterial but not for pharmaceutical purposes [167]. CBHs are promising in biomedicine fields due to their unique anti-bacterial, anti-fungal, and wound healing-promoting effects, as well as their good biocompatibility and biodegradability. The current productized goods are mainly used in health care, load medicine, antimicrobials, anti-inflammatory drugs, etc., with some of the available biomedical CBHs online shown in Table 2. There is a tremendous market for CBHs, and they deserve industrialization, considering the potential biomedical applications, as illustrated above, have been well-researched.

## 6. Conclusions and Prospects

CBHs are affordable biomaterials that are made using either physical or chemical methods. Additionally, they are biodegradable and biocompatible. They differ from other varieties of hydrogels in a variety of respects, such as simplicity of modification, low toxicity, and wide availability. Additionally, they can be blended to produce composite hydrogels, which have improved mechanical strength, stability, responsiveness, and functional properties, by adding other organic or inorganic polymers or chemicals. Further investigation into methods to protect these benign groups during synthesis is warranted because the free amino groups of CBHs and chitosan both function in a variety of synthesis processes and are essential for pH sensitivity, antimicrobial activity, and other biological reactions, among other things.

During the characterization process, most of the properties of CBHs are well-represented by the above methods of characterizing CBHs, but there are also problems that cannot be ignored. In addition to these commonly used characterization methods, the vast majority ignore the size of metabolites of CBHs in humans, how they are degraded, whether they can be eliminated from the body, and whether they are harmful to humans. The widely used antibacterial properties usually rely on *Staphylococcus aureus* and *Escherichia coli* to characterize them while ignoring fungi and other pathogenic bacteria, which deserve to be explored in depth in subsequent studies.

Versatile CBHs are proven to have great impacts on biomedical applications in laboratories, including tissue engineering, drug delivery vehicles, wound healing, BAUs, biosensors, etc. Nevertheless, only a limited quantity of CBHs have been approved for application, with only a fraction of them exploited as biomedical products. Therefore, it is necessary to improve the safety test criteria for biomedical CBH products. The promising application of smart hydrogels in various biomedical sectors is made possible by their regulated biodegradation and kinetics. But there are already some common obstacles that must be overcome, such as poor mechanical strength and restricted stability. Furthermore, dynamic covalent hydrogels are an emerging type of smart hydrogel because of their superior ability to switch forces. Ultimately, the relationship between the properties of CBHs and their biomedical applications is a comprehensive issue and turns out to be a complicated optimization problem. Many studies succeed in improving some features while neglecting others. Consequently, further study is indispensable to determine the latent relationships between numerous properties and resolve how to balance each property to improve system performance. Ultimately, combining CBHs with the 4DP technique is promising due to the technique’s favorability for large-scale production once chitosan-based materials are permitted for biomedical applications by the U.S. FDA in the near future.

## Figures and Tables

**Figure 1 polymers-15-02482-f001:**
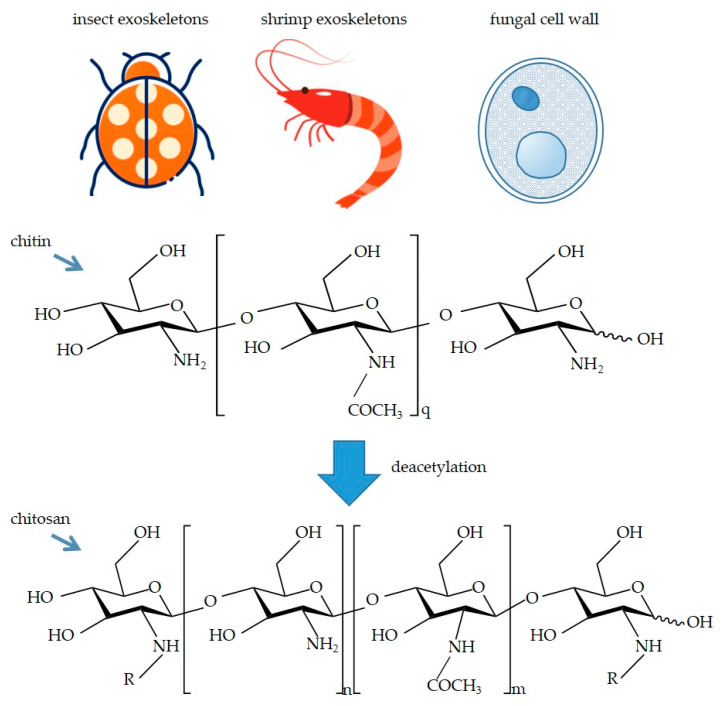
Common sources of chitosan and the structures of chitin and chitosan. -R represents acetyl group (-COMe) or -H. In addition, the degree of deacetylation is determined by the ratio of n to m.

**Figure 2 polymers-15-02482-f002:**
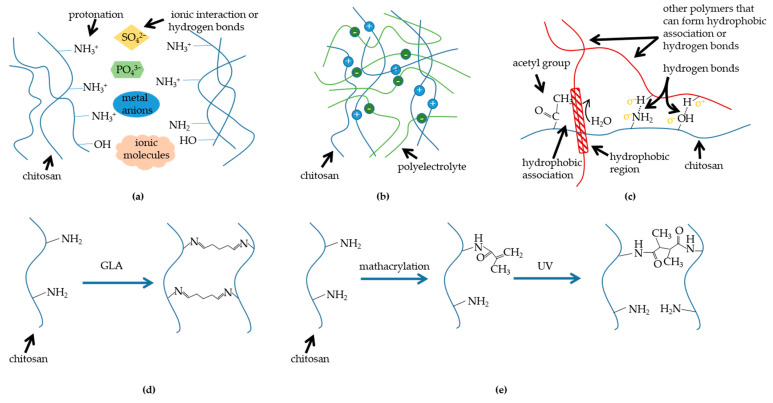
Mechanisms of diverse synthesis methods: (**a**) chitosan crosslinks with molecules; (**b**) chitosan crosslinks with polyelectrolyte; (**c**) chitosan blends with polymers which can form hydrogen bonds; (**d**) chitosan chemical crosslinker crosslinking (taking glutaraldehyde (GLA) as an example); (**e**) methacrylation and irradiation crosslinking of chitosan (mathacrylation is a kind of graft).

**Figure 3 polymers-15-02482-f003:**
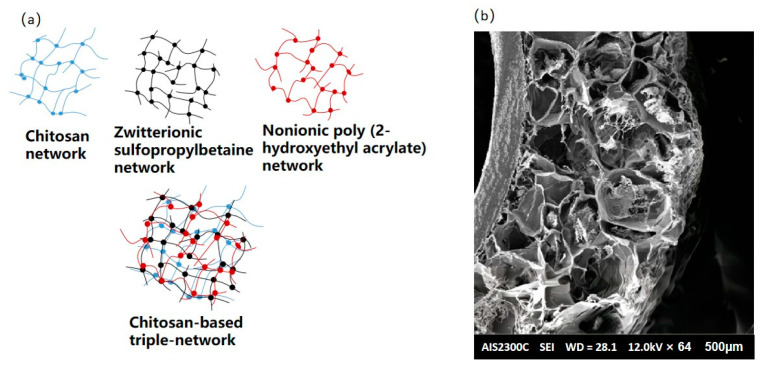
(**a**) Schematic diagram of chitosan-based triple-network hydrogel; (**b**) SEM image of CBHs combining electrospinning [42].

**Figure 4 polymers-15-02482-f004:**
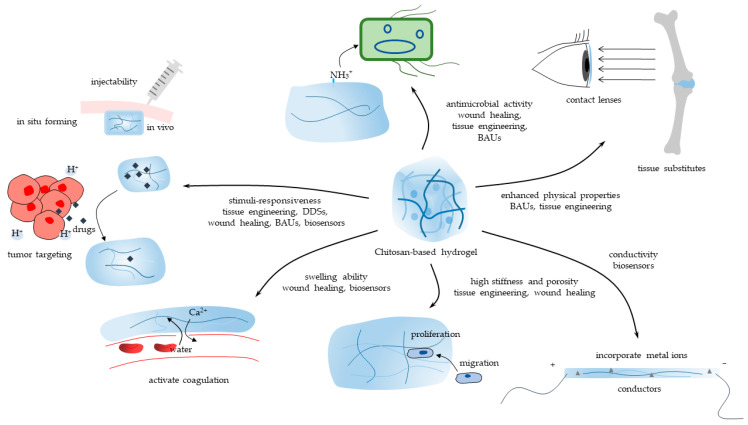
The main properties of CBHs and their biomedical application sectors.

**Table 1 polymers-15-02482-t001:** Immobilization matrix in biomedicine.

CBH Formation	Immobilized Substances	Detection Target	Detection Parameter	Low Limit of Detection	Possible Biomedical Application	References
membrane	bioprobes	Vascular endothelial growth factor (VEGF)	fluorescence signals	23 pM	diagnosis of early cancer and other diseases that involve angiogenesis	[155]
membrane	bioprobes	β-glucuronidase of *E. coil*	blue color	100 nM	diagnosis of *E. coli* infections	[156]
membrane	detection target	analytes that can interact with streptavidin	changes in the angle of light reflected by the film	1.9 × 10^−6^ RIU	biomolecular detection	[153]
membrane	bioprobes	Micro Ribonucleic acid (miRNA)	fluorescence signals	0.03 fM	detection of miRNA-21 in MCF-7 cancer cells and multicolor imaging of the cells	[149]
membrane	bioprobes	Glucose	fluorescence signals	0.029 mM	glucose monitoring in vivo	[157]
surface layer	enzymes	Glucose	current change	0.25 mM	continuous glucose monitoring systems (CGMS)	[150]
surface layer	enzymes	hydrogen peroxide	current change	0.07 mM	detection of the catalase activity in biological samples	[151]
surface layer	bioprobes	*E. coil*	raman signal	3.46 CFU/mL	hygiene	[154]
surface layer	antibodies	Interleukin-6	current change	0.03 pg/mL	detection of sepsis	[158]
surface layer	enzyme	Glucose	current change	0.1 mM	monitoring glucose levels in diabetic patients without invasive blood sampling	[152]
3D structure	bioprobes	Cu^2+^	fluorescence signals	4.75 μM	screening infectious diseases, chronic disease treatment, health management, and well-being surveillance	[159]

**Table 2 polymers-15-02482-t002:** Partial approved biomedical CBH products online.

Name of Products	Producers	Application Methods	Country
silicone of quaternary ammonium salt of chitosan	Tianjin Jiashitang Technology Co., Ltd. (Tianjin, China)	Apply the product to the scar and gently massage it three to five times a day for 3–6 months, which can effectively remove the scar.	China
Wound antibacterial spray	Henan Huibo Medical Co., Ltd. (Nanyang, China)	It is used for antibacterial nursing of postoperative wounds, pressure sores, burns and scalds to promote healing.	China
Chitosan vaginal packing dressing	Harbin Ganbaina Biological Pharmaceutical Co., Ltd. (Harbin, China)	Anti-bacterial anti-inflammatory, postoperative repair, nourishing lubrication, relieving erosion, regulating micro-ecology, tightening recovery	China
Chitosan gynecological gel	Guangzhou Lars Medical Technology Co., Ltd. (Guangzhhou, China)	For bacterial vaginosis, fungal vaginitis, increased vaginal secretions and vulva congestion swelling	China
Chitosan anorectal antibacterial gel	Shenyang Dekang Pharmaceutical Technology Co., Ltd. (Shenyang, China)	It is used to improve the symptoms of hemorrhoids bleeding, anal bulge, prolapse of hemorrhoids, congestion and edema of hemorrhoids mucosa and reduce hemorrhoids caused by internal hemorrhoids and mixed hemorrhoids.	China
NOW Chitosan capsule	NOWFOODS (Bloomingdale, IL, USA)	Decomposition of greasy, fat pressure health, oil adsorption, intestinal drainage oil	USA

## Data Availability

No new data were created or analyzed in this study.

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
