# Peer review of "Synthesis, Characterization, Properties, and Biomedical Application of Chitosan-Based Hydrogels"

_polymers, 2023, doi:10.3390/polym15112482_

Round 1

Reviewer 1 Report

The manuscript is a review on the obtaining, characterization, and uses of chitosan-based hydrogels. I found the manuscript professionally written and w organized. I have only a couple of comments on the current version.

1. Fig 1, Correct the typo in Fig 1 “graftc copolymerization” it should be graft

2. The weakness of the manuscript is on Section 3, It will be useful, since this is a review, to provide examples on how the mentioned characterization technique has been used to obtain information of chitosan hydrogels or their components, rather than explaining the fundamental of the technique.  I think, by providing more relevant examples on the use of each technique will be more interesting for the readers. I suggest removing the basis of the technique (sections 3.1 and 3.2) and include more examples of works where the techniques have been useful for determining specific features of chitosan-based hydrogels. Additionally, microbiological activity is more an application rather than an analytical technique.

Author Response

Dear reviewer,

We are appreciative of the chance to modify our paper, “Synthesis, characterization, properties, and biomedical application of chitosan-based hydrogels,” in light of the reviewers' criticisms and recommendations. After carefully examining each point raised by the editor and reviewers, the paper has been amended as necessary. The revised draft and a copy that has been annotated to show the changes made are also included. The following is a detailed answer to each criticism made by reviewers.

Point 1: Fig 1, Correct the typo in Fig 1 “graftc copolymerization” it should be graft.

Response 1: It’s an error we overlooked when drawing the figures. And we checked and corrected errors of literal contents in all figures to make them correct, brief, and precise.

Point 2: The weakness of the manuscript is on Section 3, It will be useful, since this is a review, to provide examples on how the mentioned characterization technique has been used to obtain information of chitosan hydrogels or their components, rather than explaining the fundamental of the technique.  I think, by providing more relevant examples on the use of each technique will be more interesting for the readers. I suggest removing the basis of the technique (sections 3.1 and 3.2) and include more examples of works where the techniques have been useful for determining specific features of chitosan-based hydrogels. Additionally, microbiological activity is more an application rather than an analytical technique.

Response 2: We realized that and have revised the entire section to make it more academic. We removed the basis of the technique in sections 3.1 and 3.2 and provided more relevant examples of the use of each technique. Besides, we characterized the antimicrobial properties of CBHs in more detail by comparing different methods for detecting antimicrobial properties.

The outline of the article is arranged to provide the whole procedure, from synthesis (for fabricating the CBHs) to characterization (for testing the properties) to properties and the associated biomedical applications (the core of the review). It’s acknowledged that microbiological activity is more relative to the application. In the article, the third part is about illustrating the methods to lay the groundwork. Therefore, this part is to summarize the characterization methods, facilitating the latter part. And we have revised the structure of the narration by deleting the redundant analysis mechanisms and adding more examples to make it more suitable for our goal.

We hope the revised version meets your requirements.

Sincerely yours,

Yeshun Zhang

Reviewer 2 Report

In this Manuscript entitled “Synthesis, characterization, properties, and biomedical application of chitosan-based hydrogels”, the authors provided a comprehensive assessment of the chitosan-based hydrogels (CBHs) with a focus on the affiliation between particular properties and their application domains. Additionally, this review also covered the main obstacles and future prospects of CBHs for biomedical applications. The authors concluded that CBHs have garnered significant interest in biomedical disciplines such as tissue engineering, wound healing, drug delivery, and bio-sensing applications.

Overall, some major issues are associated with this review, which need to be addressed before possible publication.

Please find the attached annotated file to see my comments.

In the end, I would like to say “Polymers journal publishes high-quality review articles related to biomedical hydrogels. Based on my comments mentioned in the annotated file, the recommendation is Major Revision.

Author Response

Dear reviewer,

We are appreciative of the chance to modify our paper, “Synthesis, characterization, properties, and biomedical application of chitosan-based hydrogels,” in light of the reviewers' criticisms and recommendations. After carefully examining each point raised by the editor and reviewers, the paper has been amended as necessary. The revised draft and a copy that has been annotated to show the changes made are also included. The following is a detailed answer to each criticism made by you.

Point 1: Rewrite this sentence again to improve the clarity. (Line 18 (in the raw PDF version you submitted))

Response 1: The structure of the raw sentence is complicated and may be confusing. "To a great extent” was meant to mean “the processing procedure could be tailored to obtain specific features." So, we revised it into “The qualities of CBHs might be greatly influenced by tailoring the manufacturing method to get certain traits, including porosity, swelling, mechanical strength, and bioactivity.” to make the structure understandable and not misleading.

Point 2: Herein? Not especially. (Line 20)

Point 3: Only biomedical applications, as provided in title?? (Line 22)

Point 4: Figures related to mechanical properties section needs to be incorporated from the literature for different mechanical properties such as SEM, TEM, (Line 27)

Point 5: Due to a large number of abbreviations used in this review. Provide a list of abbreviations to make it easier for readers to study this intriguing review. (Line 32)

Point 6: Write in subscript form?? (Line 33)

Point 7: Mean drug delivery? (Line 40)

Point 8: Rewrite the novelty of your review?? How your review would be beneficial and different from other similar reviews. (Line 44)

Response 8: We summarized the studies concerning the possible benefits of CBHs in biomedicine. This review not only introduces the synthesis and characterization methods of CBHs, but also links the properties of CBHs with their biomedical applications. Besides, development prospects, future perspectives, and existing obstacles are involved.

Point 9: Is Figure 1 is drawn with the help of [7, 8] reference??

If it is taken from [7, 8], provide copyright statement?? (Line 48)

Point 10: Schematic illustration of physical crosslinking would be beneficial for the understanding of readers. (Line 70)

Point 11: PVA is abbreviation of?

Mention all the abbreviations at the first mention? (Line 94)

Point 12: Include schematic illustration of blending, in-situ and grafting-onto-method, which are used to synthesis CBHs. (Line 131)

Point 13: Provide full form of the abbreviation at the first mention only. (Line 136)

Point 14: Gel?? Mean gelatin?? (Line 137)

Point 15: Add copyright statement?? (Line 141)

Point 16: Provide some examples about the microstructural behavior of CBHs developed for biomedical applications. (Line 142)

Point 17: This is the general information about SEM and looks ordinary, The authors should consider removing this?? (Line 155)

Point 18: CS for chitosan? but not defined at the first mention?? (Line 172)

Point 19: The authors started talking about smart hydrogels without defining it.

The authors should provide separate section about smart hydrogels used to develop scaffolds using modern manufacturing 3D/4D printing techniques.

For more information, the authors are advised to check and refer the following articles.

4D printing: technological and manufacturing renaissance. Macromolecular Materials and Engineering 307.8 (2022): 2200003.

4D printing of shape memory polymer composites: A review on fabrication techniques, applications, and future perspectives. Journal of Manufacturing Processes 81 (2022): 759-797. (Line 172)

Point 20: Common information and donot require to be included here. Instead, discuss chemical interaction analysis of CBHs. (Line 184)

Point 21: OD means?? (Line 223)

Point 22: Where are the references of this subsection?? (Line 232)

Point 23: Tabulate the antimicrobial properties and other properties of CBHs developed for different biomedical applications. (Line 240)

Point 24: write only SEM, (Line 244)

Point 25: Thermo-responsive?? (Line 252)

Point 26: The authors are advised to incorporate the following studies related to chitosan-based hydrogels:

Biopolymeric sustainable materials and their emerging applications. Journal of Environmental Chemical Engineering (2022): 108159.

Novel biopolymer-based sustainable composites for food packaging applications: A narrative review. Food Packaging and Shelf Life 33 (2022): 100892 (Line 272)

Point 27: Is this figure drawn by the authors?

If not write copyright statement? (Line 279)

Point 28: This text should be in the above paragraph instead of in Figure 3 caption. (Line 279)

Point 29: Incorporate the following article related to hydrogels used in tissue regeneration applications:

A review on four-dimensional bioprinting in pursuit of advanced tissue engineering applications. Bioprinting (2022): e00203. (Line 288)

Point 30: The article should incorporate the following articles realted to the scaffolds prepared through modern 3D printing/bioprinting:

Additive manufacturing of sustainable biomaterials for biomedical applications. Asian Journal of Pharmaceutical Sciences (2023): 100812.

Recent advances in 3D-printed polylactide and polycaprolactone-based biomaterials for tissue engineering applications. International Journal of Biological Macromolecules (2022). (Line 297)

Point 31: 3D bioprinting of thermo-responsive materials or other stimuli-responsive materials is called 4D bioprinting. The authors should include a few paragraphs above the progress of 4D bioprinted CBHs. For more information, check,

4D bioprinting of smart polymers for biomedical applications: Recent progress, challenges, and future perspectives. Reactive and Functional Polymers (2022): 105374. (Line 316)

Point 32: TFNA means?? (Line 329)

Point 33: DNA is abbreviation of?? (Line 330)

Point 35: Write in subscript. (Line 370)

Point 36: Write in subscript. (Line 371)

Point 37: It would be better to incorporate 4D printing of CBHs which is suitable for drug release applications. (Line 412)

Point 38: The authors should incorporate the following study in their article:

Additive manufacturing of sustainable biomaterials for biomedical applications. Asian Journal of Pharmaceutical Sciences (2023): 100812. (Line 414)

Point 39: DCC?? (Line 477)

Point 40: UCST?? (Line 487)

Point 41: Font size is changed. May be these lines are plagiarized. That's why font is different. (Line 494)

Point 42: in subscript? (+2) (Line 522)

Point 43: AgNPs?? (Line 541)

Point 44: iNOS?? (Line 551)

Point 45: TNF?? (Line 556)

Point 46: VEGF?? Write abbreviations or provide full list of abbreviations (Line 138)

Point 47: The authors should also incorporate the use of soft and smart CBH in developing actuators using 4D printing. For more information, check

4D printing: Technological developments in robotics applications. Sensors and Actuators A: Physical (2022): 113670. (Line 651)

Point 48: Instead of conclusions, the authors should introduce current challenges and future perspective, which must be thought-provoking and write concluding remark in the final paragraph. (Line 677)

Point 49: Write concluding remarks in the final paragraph (Line 696)

Sincerely yours,

Yeshun Zhang

Reviewer 3 Report

This current review aims for synthesis methods, property characterization, and applications of CBHs in biomedicine. This research is under the scope of this journal; the topic is relevant for readers, and this research deals with potentially significant knowledge of the field. And It will be important for Biomedicine knowledge. The topic is relevant for readers and this study deals with potentially considerable knowledge of the field and opens new ways for future studies. 

However, there are Major aspects which are needed to be improved in the manuscript:

(Keywords)

  • Please add keywords, and order the keywords / Mesh terms alphabetically

(Discussion) 

  • It was investigated in an animal study the usage of chitosan scaffolds with very good biocompatibility for dental regeneration (Dentistry). Please, read this article,  (DOI: 10.1016/j.joen.2017.03.005) investigated in an animal study the usage of this lyophilized hydrogel Chitosan scaffolds for dental regeneration. It was studying the Lyophilized gel Chitosan scaffold in the dentistry application in the pulp regeneration,  was added inside the root canal dentine walls to see recovery of dental tissues, after 3 months. This was added inside the root canal dentine walls to see the recovery of dental tissues and bone tissue. After 3 months the CS scaffold maintains in the dental canal.

  • Improve the resolution quality of all figures and graphs (and a presentation). The font/ language in the figure/caption differs from the text. Please, standardized the size and the font in the figures and charts with the font of the manuscript. 
  • Tissue engineering “porosity arrangement supports cell adhesion, proliferation, differentiation, and function”  The importance of the porosity or the space provision, please read this article, ( (2010). New formulations for space provision and bone regeneration. Biodental Eng. I1, 71-76. WOS:000282776500012; SBN 978-0-415-57394-8.) reported the influence of different formulations of hydrogel in providing an adequate porosity of the scaffold, thus emphasizing the importance of the three-dimensional distribution of particles and also space provision for new bone formation.
  •  

Author Response

Dear reviewer,

We are appreciative of the chance to modify our paper, “Synthesis, characterization, properties, and biomedical application of chitosan-based hydrogels,” in light of the reviewers' criticisms and recommendations. After carefully examining each point raised by the editor and reviewers, the paper has been amended as necessary. The revised draft and a copy that has been annotated to show the changes made are also included. The following is a detailed answer to each criticism made by you.

Point 1: (Keywords) Please add keywords, and order the keywords / Mesh terms alphabetically

Response 1: We have added “properties” as one key word because it has been mentioned in the context many times. We have arranged the keywords alphabetically to make them orderly.

Point 2: (Discussion) It was investigated in an animal study the usage of chitosan scaffolds with very good biocompatibility for dental regeneration (Dentistry). Please, read this article, (DOI: 10.1016/j.joen.2017.03.005) investigated in an animal study the usage of this lyophilized hydrogel Chitosan scaffolds for dental regeneration. It was studying the Lyophilized gel Chitosan scaffold in the dentistry application in the pulp regeneration, was added inside the root canal dentine walls to see recovery of dental tissues, after 3 months. This was added inside the root canal dentine walls to see the recovery of dental tissues and bone tissue. After 3 months the CS scaffold maintains in the dental canal.

Response 2: The unsatisfactory results were not encompassed in the post-version. The article, "Histologic evaluation of regenerative endodontic procedures with the use of chitosan scaffolds in immature dog teeth with apical periodontitis. Journal of Endodontics 2017, 43, 1279-1287.”, enlightened us that not all CBHs could be utilized directly for versatile applications. We have given our own assumptions for the reason. And finally, draw the conclusion that “the correlation between functional ingredients or parameters of CBHs and their potential biomedical applications deserves further research." Besides, the added illustration not only makes the review objective but also makes the conclusions in “5. Conclusions and prospects” profound and understandable.

Point 3: (Discussion) Improve the resolution quality of all figures and graphs (and a presentation). The font/ language in the figure/caption differs from the text. Please, standardized the size and the font in the figures and charts with the font of the manuscript.

Response 3: We have revised all possible errors in formations and contents in figures and charts.

Point 4: (Discussion) Tissue engineering “porosity arrangement supports cell adhesion, proliferation, differentiation, and function” The importance of the porosity or the space provision, please read this article, ((2010). New formulations for space provision and bone regeneration. Biodental Eng. I, 1, 71-76. WOS:000282776500012; SBN 978-0-415-57394-8.) reported the influence of different formulations of hydrogel in providing an adequate porosity of the scaffold, thus emphasizing the importance of the three-dimensional distribution of particles and also space provision for new bone formation.

Response 4: We have cited the “New formulations for space provision and bone regeneration. Biodental Eng. I, 1, 71-76. WOS:000282776500012; SBN 978-0-415-57394-8.)” to support our statement “porosity arrangement supports cell adhesion, proliferation, differentiation, and function” with “Besides, surface modification with other substances can foster the step.”

(the referred sentence is “Additionally, there is a trend in developing biologic modalities that may enhance bone healing of specific sites. With this regard, a synthetic cell-binding peptide (P-15) incorporated in a scaffold (Anorganic Bovine Matrix [ABM]) has been frequently used to facilitate the attachment, migration, and differentiation of osteoblastic cells”).”

Sincerely yours,

Yeshun Zhang

Reviewer 4 Report

Ye et al presented a review manuscript titled, "Synthesis, characterization, properties, and biomedical application of chitosan-based hydrogels". This manuscript presented the usefulness of chitosan polymer. This manuscript is well organized, well presented, but needs major revisions as mentioned below:-

1. Plagiarised texts found in all subsections of 3 except the subsection 3.3. Hence, authors must revise them.

2. Recently approved drugs employing chitosan-based hydrogels must be presented in a table format and that is to be included with recent literature reports with an explanation in 1-2 paragraphs before the conclusion section.

Author Response

Dear reviewer,

We are appreciative of the chance to modify our paper, “Synthesis, characterization, properties, and biomedical application of chitosan-based hydrogels,” in light of the reviewers' criticisms and recommendations. After carefully examining each point raised by the editor and reviewers, the paper has been amended as necessary. The revised draft and a copy that has been annotated to show the changes made are also included. The following is a detailed answer to each criticism made by you.

Point 1: Plagiarised texts found in all subsections of 3 except the subsection 3.3. Hence, authors must revise them.

Response 1: In the subsections 3, more specific explanations have been expressed via the authors’ own words and the references that support the statements have been added.

The outline of the article is arranged to provide the whole procedure, from synthesis (for fabricating the CBHs) to characterization (for testing the properties) to properties and the associated biomedical applications (the core of the review). And we have revised the structure of the narration of Section 3 by deleting the redundant analysis mechanisms and adding more examples to make it more suitable for our goal.

Point 2: Recently approved drugs employing chitosan-based hydrogels must be presented in a table format and that is to be included with recent literature reports with an explanation in 1-2 paragraphs before the conclusion section.

Response 2: We added the new section “5. The status of approved applications of CBH in biomedicine” and illustrated the relative information based on the limited intel online, considering many products are not available online. What’s more, there are few papers providing the approval process for their CBHs. Therefore, it’s admitted that the section may provide lame perspectives. Besides, chitosan-based materials are still not permitted by the U.S. FDA, which may hinder the ratification process. But we hope it could be useful to illustrate the application status to some extent.

Sincerely yours,

Yeshun Zhang

Round 2

Reviewer 2 Report

The authors have addressed all of the comments, and the article is in acceptable form, now.

Minor spell checks are required. 

Reviewer 3 Report

The authors improve the article with the reviewer’s comments. 

Reviewer 4 Report

The authors made the suggested corrections.

The manuscript can be accepted in its present form.